# Prevalence and associated factors of stunting among under-five children in Ethiopia: Application of marginal models analysis of 2016 Ethiopian demographic and health survey data

**Woldemariam Erkalo Gobena**[1]*, **Teramaj Wongel Wotale**[1], **Mesfin Esayas Lelisho**[2], **Wubishet Gezimu**[3]*

1 Department of Statistics, College of Natural Science, Mettu University, Mettu, Ethiopia, 2 Department of Statistics, College of Natural Science, Mizan-Tepi University, Tepi, Ethiopia, 3 Department of Nursing, College of Health Sciences, Mettu University, Mettu, Ethiopia

* wubishet151@gmail.com (WG); woldenegniko@gmail.com (WEG)

**Data Availability Statement:** Third party data was obtained for this study from the DHS Program.

## Abstract

### Background

Stunting, short for age, affects the overall growth and development of the children. It occurs due to chronic under nutrition. Stunting vastly occurs in impoverished regions of the world, including Ethiopia.

### Objective

This study aimed to investigate the prevalence and correlates of stunting among under-five children in Ethiopia using marginal models.

### Methods

Data were taken from the 2016 Ethiopian Demographic Health Survey, which is a nationally representative survey of children in the 0–59 month age group. For marginal models, generalized estimating equations and alternating logistic regression models were used for the analysis.

### Results

The prevalence of stunting among the under-five children was 34.91% in the area. The proportion was slightly higher among male (36.01%) than female (33.76%) child. The Alternating Logistic Regression model analysis revealed that the child's age, the mother's education level, the mother's body mass index, the place of residence, the wealth index, and the previous birth interval were found to be significant determinants of childhood stunting, and the result shows that children born with a lower previous birth interval (less than 24 months)

Data may be requested from the DHS Program after creating an account. More access information can be found on The DHS Program website (https://dhsprogram.com/data/Access-Instructions.cfm). The authors confirm that interested researchers would be able to access these data in the same manner as the authors. The authors also confirm that they had no special access privileges that others would not have.

**Funding:** The author(s) received no specific funding for this work.

**Competing interests:** The authors have declared that no competing interests exist

**Abbreviations:** AIC, Akaike's information criterion; BMI, Body Mass Index; CSA, Central Statistical Agency; EDHS, Ethiopian Demographic and Health Survey; WHO, World Health Organization.

were more likely to be stunted than those born within a higher birth interval. Children in rural Ethiopia were more likely to be stunted than children in urban Ethiopia.

## Conclusion

This study found that more than one third of children were stunted in the area. The study also determined that child's age, the mother's education, the mother's body mass index, the place of residence, the wealth index, and birth interval influence stunting. Therefore, it is better enhancing the nutritional intervention programs.

## Introduction

Childhood malnutrition is the consequence of inadequate food intake, illnesses including diarrhea and other gastrointestinal illnesses, poor hygiene, and uneducated parents [1, 2]. Child malnutrition antecedents food insecurity, poor perinatal care, poor healthcare services, and environment factors [3]. As the result it bears noticeable detrimental effects on the body function [4]. It is the most common anthropometric impairment noticed among under-five children in the world's poorest regions [5, 6]. It accounts for about half of childhood mortality, typically in resource-limited settings [7].

Stunting is a type of child malnutrition that is characterized by being short for one's age. When the measured values of the height-for-age z-score are more than two standard deviations below the typical World Health Organization (WHO) Child Growth Standards median, it is diagnosed anthropometrically [8]. Around 149 million under-five children were stunted globally. Although the proportion of stunting has declined in sub-Saharan Africa, the least developed region of the world, it still remains at over 30% [9]. In Ethiopia, a nation in sub-Saharan Africa, the prevalence of stunting has significantly dropped. However, the figure is still higher in some regions of the country. For instance, in the Afar region, it was about 41%, which is above the national average. Moreover, the figure was higher in the Amhara region, Benishangul-Gumuz region, and Dire Dawa City Administration [10, 11].

Certain biological and socio-demographic factors have been linked to stunting. Sex and the child's age were found to be two biologic factors that affect stunting. A male child was more likely than a female to be stunted [12–14]. Stunting has been linked to maternal education, antenatal (ANC) visits during pregnancy, and a history of bottle feeding [12, 15, 16]. A study carried out in Kenya and Cambodia revealed an association between child stunting and residency and family income [14, 16]. Additionally, it is associated to illnesses like chronic gastroenteritis, anemia, and parasite infection [13, 17, 18].

The proportion of stunting has reduced in Ethiopia since 2000, but it is still too slow to meet the WHO's global target of reducing stunting to 40% by 2025 [19]. Several African nations, including Ethiopia, modified their national nutrition policies, strategies, and action plans to combat child malnutrition, including stunting [20].

The majority of previous studies in Ethiopia that used EDHS data ignored the appropriate sampling design and the clustering nature of the EDHS data and examined factors contributing to childhood stunting. This study, therefore, aimed to fill this gap by applying the marginal model analysis to identify factors associated with stunting among under-five children in Ethiopia using a stratified, two-stage cluster sampling design.

## Methods and materials

### Data sources

For this study, the 2016 EDHS sample was selected using a stratified, two-stage cluster sampling design and house-to-house cross-sectional survey data extracted from the 2016 EDHS were used. The data were collected by Ethiopia's Central Statistical Agency (CSA) between January 18 and June 27, 2016. Like all observational studies, this one adhered to the "Strengthening the reporting of observational studies in epidemiology (StroBE) statement: guidelines for reporting observational studies." (S1 Checklist).

### Study population and variables

In the households of the selected clusters, a total of 10641 children under the age of 59 months were found. The complete height-for-age record for 4621 of the children and 626 clusters were gathered from them. Stunting is assessed in this study using the height-for-age Z-score (multiplied by 100), wh1ich is utilized as a response variable. Children are termed stunted if their height-for-age Z-score is below minus two standard deviations (-2 SD) from the median, as opposed to children whose height-for-age Z-score is above minus two standard deviations (-2 SD) from the median [5].

As proxy indicators of socioeconomic, demographic, health, and environmental characteristics, the following fourteen factors were included: the mother's education level, wealth index, number of household members, place of residence, age of the child, sex of the child, mother's age at birth of the child, number of antenatal care visits, previous birth interval, birth order, mother's body mass index, diarrhea, fever, and source of drinking water.

### Statistical analysis

Descriptive analysis and marginal models were applied to achieve the main purpose of the study and draw significant conclusions.

**Marginal models.** Marginal models are statistical models widely used to model clustered or repeated data. The principal objective of the marginal models is to analyze the population-averaged effects of the given predictor on the binary response variable of interest. This means that the covariates are directly related to the marginal expectations [21]. The marginal models used in the study were the Generalized Estimating Equation (GEE) and Alternating Logistic Regression (ALR).

**Generalized estimating equations (GEE).** GEE uses correlation to measure the association within clusters or subjects in terms of marginal correlations [21]. It is used to account for the correlation between responses of interest for subjects in the same cluster [22]. For clustered as well as repeated measured data, Liang and Zeger [23] projected GEE, which requires only the correct specification of the correlation structure. The GEE model, based on a generalized linear model, and the working correlation structure are given by:

$$g\left(\pi_j\right) = logit\left(\pi_j\right) = X'_j\beta$$

where $g(\pi_j)$ is logit link function, $X_j$ is $(n_j \times \rho)$ dimensional vector of known covariates, $\beta$ is (1 x p) dimensional vector of the unknown fixed regression parameter to be estimated, $E(Y_j) = \pi_j$ is expected value of the responses $Y_j$ in $j^{th}$ cluster and $Y_j$ is distributed binomially as $Y_j \sim Bin(n_j, \pi_j)$.

**Parameter estimation for GEE.** GEE follows the quasi-likelihood estimation method and estimates parameter $\hat{\beta}$ by solving estimating equations that consist of the working correlation matrix ($R_j$) and matrix with marginal variances on the main diagonal and zeros elsewhere

($A_j$). The score equation used to estimate the marginal regression parameters while accounting for the correlation structure is given by:

$$S(\beta) = \sum\nolimits_{j=1}^{m} \frac{\partial \pi_j}{\partial \beta'} \left[ A_j^{\frac{1}{2}} R_j A_j^{\frac{1}{2}} \right]^{-1} \left( Y_j - \pi_j \right) = 0$$

## Alternating logistic regression model (ALR)

ALR differs from GEE in that ALR measures the association among the observed data using an odds ratio, whereas GEE measures this association through the correlation structure. ALR is an extension of GEE in the sense that the precision estimates follow both regression and association parameters, $\beta$ and $\alpha$, respectively. Moreover, inferences can be made about marginal parameters and pairwise associations between subjects using ALR [21].

Let $\gamma_{jkl}$ be the log odds ratio between outcomes $Y_{jk}$ and $Y_{jl}$ and let $\pi_{jk} = P(Y_{jk} = 1)$ and then the association of the two responses is defined as [21]:

$$logit\, P\left( Y_{jk} = 1/Y_{jl} = y_{jl} \right) = \gamma_{jkl} y_{jsl} + \log\left( \frac{\pi_{jk} - v_{jkl}}{1 - \pi_{jk} - \pi_{jl} + v_{jkl}} \right)$$

Assume $\gamma_{jkl} = \alpha$ Then the pair-wise log odds ratio $\alpha$ is the regression coefficient in logistic regression of $Y_{jk}$ on $Y_{jl}$.

## Parameter estimation of ALR

ALR follows quasi-likelihood estimation approach like GEE. Let $\xi_j$ be a vector with elements $\xi_{jkl} = E(Y_{jk}/Y_{jl} = y_{jl})$ and $R_j$ be the vector of residual with elements $R_{jkl} = Y_{jk} - E(Y_{jk}/Y_{jl} = y_{jl}) = Y_{jk} - \mu_{jkl}$. Let $S_j$ be a vector of diagonal matrix with diagonal element $\xi_{jkl}(1 - \xi_{jkl})$ and let $W_j$ be matrix $\frac{\partial \xi_j}{\partial \alpha}$. Finally, let $A_j = Y_j - \pi_j$, $B_j = cov(Y_j)$, and $C_j = \frac{\partial \pi_j}{\partial \beta}$. Then the alternating logistic regression parameter $\delta = (\beta, \alpha)$ is the simultaneous solution of the following unbiased estimating equations [23].

$$U_\beta = \sum\nolimits_{j=1}^{m} C_j' B_j^{-1} A_j = 0$$

$$U_\alpha = \sum\nolimits_{j=1}^{m} W_j' S_j^{-1} R_j = 0$$

The above unbiased estimating equations are used to solve $\beta$ and $\alpha$ using Gauss-Seidel procedure algorithm.

## Results

### Descriptive statistics

Of the 4,621 children included in this study, 50.9% were male and 49.10% were female. Around 1613 (34.91%) under-five children were stunted in the study area. Of these, about 36.01% and 33.76% were male and female, respectively (Table 1).

### Multicollinearity diagnostic test

Before building the model for determinants of stunting, a set of independent variables (predictors) must be tested for multicollinearity. The statistic used for the multicollinearity test is the variance inflation factor (VIF). For each predictor, VIF is less than 10, which means

**Table 1. Cross tabulation of prevalence of stunting by maternal, socioeconomic and demographic variables.**

| Predictors/ Independent Variables | | Stunting (%) | | |
|---|---|---|---|---|
| | | **Not-stunted** | **St Stunted** | **Total** |
| Child's sex | Female | 1,503 (66.24) | 766 (33.76) | 2,269 |
| | Male | 1,505 (63.99) | 847 (36.01) | 2,352 |
| Child's age | Less than 6 months | 626 (87.19) | 92 (12.81) | 718 |
| | 6–11 months | 598 (84.70) | 108 (15.30) | 706 |
| | 12–17 months | 493 (66.98) | 243 (33.02) | 736 |
| | 18–23 months | 279 (52.35) | 254 (47.65) | 533 |
| | 24–59 months | 1,012 (52.49) | 916 (47.51) | 1,928 |
| Previous birth interval | Less than 24 | 563 (61.20) | 357 (38.80) | 920 |
| | 24 up to 47 | 1,525 (63.94) | 860 (36.06) | 2,385 |
| | 48 or above | 920 (69.91) | 396 (30.09) | 1,316 |
| Birth order | 1up to 3 | 1,236 (69.40) | 545 (30.60) | 1,781 |
| | 4 or above | 1,772 (62.39) | 1,068 (37.61) | 2,840 |
| Mother's education | No education | 1,406 (59.53) | 956 (40.47) | 2,362 |
| | Primary | 1,033 (67.43) | 499 (32.57) | 1,532 |
| | Secondary or above | 569 (78.27) | 158 (21.73) | 727 |
| Number of household member | 1up to 4 | 666 (68.87) | 301 (31.13) | 967 |
| | 5 up to 9 | 2,181 (64.11) | 1,221 (35.89) | 3,402 |
| | 10 or above | 161 (63.89) | 91 (36.11) | 252 |
| Place of residence | Rural | 2,425 (62.69) | 1,443 (37.31) | 3,868 |
| | Urban | 583 (77.42) | 170 (22.58) | 753 |
| Age of mother | 11–19 years | 1,964 (64.25) | 1,093 (35.75) | 3,057 |
| | 20–49 years | 1,044 (66.75) | 520 (33.25) | 1,564 |
| Mother's Body Mass Index | Thin | 632 (60.13) | 419 (39.87) | 1,051 |
| | Normal | 2,011 (65.00) | 1,083 (35.00) | 3,094 |
| | Overweight | 365 (76.68) | 111 (23.32) | 476 |
| Wealth index | Poor | 1,550 (61.90) | 954 (38.10) | 2,504 |
| | Medium | 503 (65.92) | 260 (34.08) | 763 |
| | Rich | 955 (70.53) | 399 (29.47) | 1,354 |
| No. of antenatal care visits | No visits | 1,036 (61.37) | 652 (38.63) | 1,688 |
| | 1–3 visits | 884 (63.60) | 506 (36.40) | 1,390 |
| | 4–20 visits | 1,088 (70.51) | 455 (29.49) | 1,543 |
| Source of drinking water | Improved water | 1,827 (65.79) | 950 (34.21) | 2,777 |
| | Unimproved water | 1,181 (64.05) | 663 (35.95) | 1,844 |
| Had diarrhea recently | No | 2,607 (65.44) | 1,377 (34.56) | 3,984 |
| | Yes | 401 (62.95) | 236 (37.05) | 637 |
| Had fever in the two weeks before survey | No | 2,544 (65.62) | 1,333 (34.38) | 3,877 |
| | Yes | 464 (62.37) | 280 (37.63) | 744 |

there is no interaction between the predictors that might affect the results of the analysis. This implies that all independent variables are fit to be used in developing the model for stunting in Ethiopia (Table 2).

## Analysis of generalized estimating equations (GEE)

Under GEE, the model-building strategy was started by fitting a model containing all possible predictors in the data. This was done by considering two different working correlation assumptions (exchangeable and independent). To select the important predictors related to

**Table 2. Multicollinearity test of independent variables.**

| Independent variables | Multicollinearity Statistics (VIF) |
|---|---|
| Child's sex | 1.00385 |
| Child's age | 1.01445 |
| Mother's age | 1.03989 |
| Mother's education level | 1.31982 |
| Mother's BMI | 1.11961 |
| Wealth index | 1.23222 |
| Birth order | 1.45688 |
| Previous birth interval | 1.06597 |
| No. of household members | 1.34526 |
| No. antenatal care visits | 1.22601 |
| Place of residence | 1.42809 |
| Water | 1.18361 |
| Had diarrhea recently | 1.10928 |
| Had fever in the two weeks before survey | 1.10777 |

stunting, the backward elimination procedure was used. The full model for the probability that the child 'i' is stunted in cluster j was fitted as

$$
\begin{aligned}
logit\left(\pi_{ij}\right) = \quad & \beta_0 + \beta_1 CSEX_M + \beta_2 CAGE_2 + \beta_3 CAGE_3 + \beta_4 CAGE_4 + \beta_5 CAGE_5 \\
& + \beta_6 MAGE_{20-49} + \beta_7 MEDU_{Pr} + \beta_8 MEDU_{Seco} + \beta_9 BMI_N + \beta_{10} BMI_{Ov} \\
& + \beta_{11} PBINT_{24-47} + \beta_{12} PBINT_{48+} + \beta_{13} NHHM_{5-9} + \beta_{14} NHHM_{10+} \\
& + \beta_{15} BORD_{4+} + \beta_{16} WINDEX_{Mi} + \beta_{17} WINDEX_{Ri} + \beta_{18} PLRESID_{Ru} \\
& + \beta_{19} ANV_{1-3} + \beta_{20} ANV_{4-20} + \beta_{21} SDW_{Un} + \beta_{22} DIAR_Y + \beta_{23} FEV_Y
\end{aligned}
$$

The subscript in each predictor are defined as,
*M = Male, 2 = 6–11, 3 = 12–17, 4 = 18–23, 5 = 24–59, Pr = Primary, Seco = Secondary or above, N = Normal, Ov = Overweight, 48+ = 48 or above, 10+ = 10 or above, 4+ = 4 or above, Mi = Middle, Ri = Rich, Ru = Rural, Un = Unimproved, Y = Yes.*

After fitting the model, predictors with insignificant p-value were ignored, and the model was refitted with the rest of the predictors sequentially. Children's sex, birth order, number of household members, age of the mother, number of antenatal care visits, source of drinking water, having ever had diarrhea, and having ever had fever were the predictors excluded from the model (Table 3).

The QIC values of the full model and the reduced model are 5362.91 (Table 3) and 5357.97 (Table 4), respectively. Then the model with the age of the child, previous birth interval, mother's education level, BMI of the mother, wealth index, and place of residence became the most preferable model. Finally, as is customary, a comparison of empirical and model-based standard errors for the parameter estimates obtained based on the given working correlation assumptions (exchangeability and independence) was performed using the selected predictors. The correlation structure in which the model-based and empirical standard errors are closest to each other is referred to as the best assumption correlation structure (Table 5). Then, from Table 4, the exchangeable working correlation assumption was found to be plausible since the two standard errors were closer to each other with the correlation parameter ($\alpha = 0.0443$). Therefore, the final proposed generalized estimating equation model for childhood stunting is

**Table 3. The full model test for variable selection in GEE.**

| Analysis of GEE Parameter Estimates Empirical Standard Error Estimates | | | | | | |
|---|---|---|---|---|---|---|
| Parameter | Estimate | S.E | 95% Confidence Limits | | Z | Pr > \|Z\| |
| Intercept | -1.9952 | 0.2145 | -2.4157 | -1.5748 | -9.30 | < .0001* |
| Sex of child | 0.0715 | 0.0682 | -0.0622 | 0.2051 | 1.05 | 0.2945 |
| Age of child | 1.1907 | 0.1473 | 0.9020 | 1.4793 | 8.09 | < .0001* |
| Age of Mother | -0.0134 | 0.0709 | -0.1523 | 0.1255 | -0.19 | 0.8499 |
| Mother's educational status | -0.2171 | 0.0802 | -0.3743 | -0.0599 | -2.71 | 0.0068* |
| BMI of Mother | -0.4032 | 0.1387 | -0.6750 | -0.1313 | -2.91 | 0.0036* |
| Birth order | -0.0056 | 0.0853 | -0.1728 | 0.1615 | -0.07 | 0.9474 |
| Previous birth interval | -0.2263 | 0.1004 | -0.4231 | -0.0295 | -2.25 | 0.0242* |
| Number of household members | 0.1388 | 0.1029 | -0.0628 | 0.3404 | 1.35 | 0.1771 |
| Number of antenatal care visits | 0.0786 | 0.0864 | -0.0907 | 0.2478 | 0.91 | 0.3630 |
| Wealth status | -0.3782 | 0.0919 | -0.5584 | -0.1981 | -4.11 | < .0001* |
| Place of residence | 0.4707 | 0.1267 | 0.2224 | 0.7190 | 3.72 | 0.0002* |
| Source of drinking water | -0.1132 | 0.0735 | -0.2573 | 0.0308 | -1.54 | 0.1235 |
| Had fever in the two weeks before survey | 0.1571 | 0.0932 | -0.0256 | 0.3398 | 1.69 | 0.0919 |
| Had diarrhea recently | 0.1128 | 0.1001 | -0.0833 | 0.3090 | 1.13 | 0.2596 |
| GEE Fit Criteria: | | | | | | |
| QIC = 5362.9173 | | | | | | |

Note: * $p < 0.05$

**Table 4. Empirical and model based standard errors for two proposed working correlation.**

| Parameter | | Analysis of GEE Parameter Estimates | | | | | |
|---|---|---|---|---|---|---|---|
| | | Estimate | S.E | 95% C.I | | Z | Pr > \|Z\| |
| Intercept | | -1.8715 | 0.1902 | -2.2443 | -1.4987 | -9.84 | < .0001* |
| Child's age | 6–11 months | 0.2380 | 0.1532 | -0.0622 | 0.5383 | 1.55 | 0.1202 |
| | 12–17 months | 1.2052 | 0.1462 | 0.9187 | 1.4917 | 8.25 | < .0001* |
| | 18–23 months | 1.8987 | 0.1445 | 1.6155 | 2.1819 | 13.14 | < .0001* |
| | 24–59 months | 1.8963 | 0.1264 | 1.6487 | 2.1440 | 15.01 | < .0001* |
| Education level | Primary | -0.2124 | 0.0797 | -0.3686 | -0.0561 | -2.66 | 0.0077* |
| | Secondary or above | -0.5659 | 0.1204 | -0.8019 | -0.3298 | -4.70 | < .0001* |
| BMI of mother | Normal | -0.1287 | 0.0796 | -0.2847 | 0.0273 | -1.62 | 0.1058 |
| | Overweight/Obese | -0.3950 | 0.1385 | -0.6665 | -0.1235 | -2.85 | 0.0043* |
| Previous birth interval | 24–47 months | -0.0816 | 0.0839 | -0.2460 | 0.0829 | -0.97 | 0.3311 |
| | 48 months or above | -0.2350 | 0.1002 | -0.4315 | -0.0385 | -2.34 | 0.0191* |
| Wealth index | Middle | -0.1671 | 0.0964 | -0.3561 | 0.0218 | -1.73 | 0.0830 |
| | Rich | -0.3681 | 0.0886 | -0.5418 | -0.1944 | -4.15 | < .0001* |
| Pace of residence | Rural | 0.4719 | 0.1199 | 0.2368 | 0.7069 | 3.93 | < .0001* |
| GEE Fit Criteria | | | | | | | |
| QIC = 5357.9710 | | | | | | | |
| Alpha (α) = 0.0443 | | | | | | | |

* $p < 0.05$ S.E: Standard Error 95% C.I: Confidence Interval

**Table 5. Empirical and model based standard errors for two proposed working correlation.**

| Coefficient | Exchangeable | | | Independent | | |
|---|---|---|---|---|---|---|
| | Estimates | Model based (S.E) | Empirical (S.E) | Estimates | Model based (S.E) | Empirical (S.E) |
| $\beta_0$ | -1.8715 | 0.1925 | 0.1902 | -1.8434 | 0.1854 | 0.1904 |
| $\beta_2$ | 0.2380 | 0.1543 | 0.1532 | 0.2337 | 0.1547 | 0.1536 |
| $\beta_3$ | 1.2052 | 0.1381 | 0.1462 | 1.2046 | 0.1382 | 0.1472 |
| $\beta_4$ | 1.8987 | 0.1440 | 0.1445 | 1.9067 | 0.1441 | 0.1458 |
| $\beta_5$ | 1.8963 | 0.1229 | 0.1264 | 1.8979 | 0.1225 | 0.1270 |
| $\beta_7$ | -0.2124 | 0.0765 | 0.0797 | -0.2224 | 0.0752 | 0.0806 |
| $\beta_8$ | -0.5659 | 0.1165 | 0.1204 | -0.6263 | 0.1143 | 0.1197 |
| $\beta_9$ | -0.1287 | 0.0795 | 0.0796 | -0.1224 | 0.0793 | 0.0787 |
| $\beta_{10}$ | -0.3950 | 0.1406 | 0.1385 | -0.4423 | 0.1407 | 0.1394 |
| $\beta_{11}$ | -0.0816 | 0.0858 | 0.0839 | -0.0672 | 0.0860 | 0.0858 |
| $\beta_{12}$ | -0.2350 | 0.0997 | 0.1002 | -0.2251 | 0.0989 | 0.1023 |
| $\beta_{16}$ | -0.1671 | 0.0959 | 0.0964 | -0.1251 | 0.0946 | 0.0964 |
| $\beta_{17}$ | -0.3681 | 0.0856 | 0.0886 | -0.3242 | 0.0820 | 0.0889 |
| $\beta_{18}$ | 0.4719 | 0.1204 | 0.1199 | 0.4312 | 0.1105 | 0.1204 |

given as:

$$logit\left(\pi_{ij}\right) = \beta_0 + \beta_2 CAGE_2 + \beta_3 CAGE_3 + \beta_4 CAGE_4 + \beta_5 CAGE_5 + \beta_7 MEDU_{Pr}$$
$$+ \beta_8 MEDU_{Seco} + \beta_9 BMI_N + \beta_{10} BMI_{Ov} + \beta_{11} PBINT_{24-47} + \beta_{12} PBINT_{48+}$$
$$+ \beta_{16} WINDEX_{Mi} + \beta_{17} WINDEX_{Ri} + \beta_{18} PLRESID_{Ru}$$

For the final GEE model, the parameter estimates and their corresponding empirically corrected standard errors with p-values are presented in Table 4.

## Analysis of Alternating Logistic Regression model (ALR)

Model building for ALR follows the same procedure as GEE's model building strategy. First, the ALR model is fitted using all the proposed predictors. Then, the insignificant predictor was removed. The child's sex, birth order, age of mother, number of antenatal care visits, number of household members, source of drinking water, having had fever in the two weeks before the survey began, and having had diarrhea recently were removed. The QIC values of the ALR models (for the full model and reduced model) are 5362.82 (Table 6) and 5357.87 (Table 7), respectively.

Therefore, the reduced ALR model was considered the best candidate model. Using the selected covariates and the association parameter, α, an alternating logistic regression (ALR) model that provides information about the pairwise association of observations between two different individuals within the same cluster was fitted. Therefore, the final proposed ALR model, including association parameters for childhood stunting is given as:

$$logit\left(\pi_{ij}\right) = \alpha + \beta_0 + \beta_2 CAGE_2 + \beta_3 CAGE_3 + \beta_4 CAGE_4 + \beta_5 CAGE_5 + \beta_7 MEDU_{Pr}$$
$$+ \beta_8 MEDU_{Seco} + \beta_9 BMI_N + \beta_{10} BMI_{Ov} + \beta_{11} PBINT_{24-47} + \beta_{12} PBINT_{48+}$$
$$+ \beta_{16} WINDEX_{Mi} + \beta_{17} WINDEX_{Ri} + \beta_{18} PLRESID_{Ru}$$

**Table 6. The full model test for variable selection in ALR.**

| Parameter | Estimate | S.E | 95% Confidence Limits | | Z | Pr > \|Z\| |
|---|---|---|---|---|---|---|
| | | | Analysis of ALR Parameter Estimates Empirical Standard Error Estimates | | | |
| Intercept | -1.9990 | 0.2142 | -2.4188 | -1.5791 | -9.33 | < .0001* |
| Sex of child | 0.0699 | 0.0682 | -0.0637 | 0.2036 | 1.03 | 0.3050 |
| Age of child | 1.1843 | 0.1471 | 0.8959 | 1.4726 | 8.05 | < .0001* |
| Age of Mother | -0.0133 | 0.0707 | -0.1518 | 0.1253 | -0.19 | 0.8511 |
| Mother's educational status | -0.2149 | 0.0801 | -0.3720 | -0.0579 | -2.68 | 0.0073* |
| BMI of Mother | -0.4017 | 0.1384 | -0.6729 | -0.1305 | -2.90 | 0.0037* |
| Birth order | -0.0025 | 0.0850 | -0.1692 | 0.1641 | -0.03 | 0.9762 |
| Previous birth interval | -0.2210 | 0.1006 | -0.4181 | -0.0239 | -2.20 | 0.0280* |
| Number of household members | 0.1423 | 0.1029 | -0.0593 | 0.3440 | 1.38 | 0.1666 |
| Number of antenatal care visits | 0.0810 | 0.0861 | -0.0878 | 0.2499 | 0.94 | 0.3469 |
| Wealth status | -0.3748 | 0.0917 | -0.5546 | -0.1950 | -4.09 | < .0001* |
| Place of residence | 0.4736 | 0.1266 | 0.2255 | 0.7218 | 3.74 | 0.0002* |
| Source of drinking water | -0.1170 | 0.0733 | -0.2606 | 0.0266 | -1.60 | 0.1103 |
| Had fever in the two weeks before survey | 0.1565 | 0.0930 | -0.0258 | 0.3388 | 1.68 | 0.0924 |
| Had diarrhea recently | 0.1119 | 0.1002 | -0.0844 | 0.3083 | 1.12 | 0.2638 |
| Alpha (α) | 0.2232 | 0.0430 | 0.1389 | 0.3074 | 5.19 | < .0001* |
| GEE Fit Criteria: | | | | | | |
| QIC = 5362.8177 | | | | | | |

Note: * $p < 0.05$

## Comparison of GEE and ALR models

In marginal models, model comparison is based on quasi-likelihood criteria (QIC). From Tables 4 and 7, the QIC values of GEE and ALR are 5357.97 and 5357.87, respectively, which are almost equal. However, the empirically corrected standard errors for the ALR model are somewhat smaller than their counterparts under the GEE model. This implies that the ALR fits the data with fewer disturbances than the GEE. Moreover, ALR extends beyond the classical GEE in the sense that precision estimates follow for both the regression parameters $\beta$ and the association parameters $\alpha$. I was also in a position to emphasize that the association is strongly significant (p < 0.0001), provided it has been correctly specified, a declaration I could not make in the corresponding exchangeable GEE analysis. Therefore, it is possible to conclude that ALR is a better model for explaining the marginal association between childhood stunting and the selected predictors. Thus, the interpretation of the parameter is based on the final proposed ALR model. Overall, the parameter estimates under ALR are slightly lower than those under GEE. This difference in parameter estimates from the two models might be due to the fact that ALR takes the associations into account, whereas GEE does not consider the association parameters in the model.

## Parameter interpretation of marginal models

Table 7 presents the parameter estimates and their corresponding empirically corrected standard errors with p values from the ALR model. Each parameter reflects the effect of the predictor on the log odds of the probability of being stunted after controlling other variables in the model. Then, the odds ratio of variables is calculated as the exponent of $\beta_j$ i.e odds ratio = $e^{\beta_j}$.

From Table 7, the results of the ALR analysis revealed that the age of the child is significantly related to stunting. After controlling all other predictors in the model, children aged

**Table 7. Parameter estimates (empirically corrected standard errors) from ALR.**

| Parameter | | Estimate | S.E | 95% C.I | | Z | Pr > \|Z\| |
|---|---|---|---|---|---|---|---|
| | | **Analysis of ALR Parameter Estimates** | | | | | |
| Intercept | | -1.8714 | 0.1900 | -2.2438 | -1.4991 | -9.85 | < .0001* |
| Child's age | 6–11 months | 0.2364 | 0.1528 | -0.0631 | 0.5358 | 1.55 | 0.1219 |
| | 12–17 months | 1.1992 | 0.1462 | 0.9128 | 1.4857 | 8.20 | < .0001* |
| | 18–23 months | 1.8930 | 0.1450 | 1.6088 | 2.1771 | 13.06 | < .0001* |
| | 24–59 months | 1.8917 | 0.1266 | 1.6435 | 2.1398 | 14.94 | < .0001* |
| Education level | Primary | -0.2109 | 0.0796 | -0.3670 | -0.0548 | -2.65 | 0.0081* |
| | Secondary or above | -0.5662 | 0.1203 | -0.8019 | -0.3304 | -4.71 | < .0001* |
| BMI of mother | Normal | -0.1300 | 0.0795 | -0.2859 | 0.0258 | -1.64 | 0.1019 |
| | Overweight/Obese | -0.3959 | 0.1383 | -0.6670 | -0.1248 | -2.86 | 0.0042* |
| Previous birth interval | 24–47 months | -0.0788 | 0.0840 | -0.2433 | 0.0858 | -0.94 | 0.3481 |
| | 48 months or above | -0.2300 | 0.1006 | -0.4272 | -0.0329 | -2.29 | 0.0222* |
| Wealth index | Middle | -0.1644 | 0.0962 | -0.3529 | 0.0241 | -1.71 | 0.0874 |
| | Rich | -0.3616 | 0.0885 | -0.5350 | -0.1883 | -4.09 | < .0001* |
| Pace of residence | Rural | 0.4727 | 0.1200 | 0.2376 | 0.7079 | 3.94 | < .0001* |
| Alpha ($\alpha$) | | 0.2114 | 0.0425 | 0.1281 | 0.2947 | 4.97 | < .0001* |
| GEE Fit Criteria | | | | | | | |
| QIC = 5357.8652 | | | | | | | |

**Note:** * $p < 0.05$; S.E: Standard Error; C.I: Confidence Interval

18–23 months were exp(1.8930) = 6.6393 (95% CI: 4.9968, 8.8207) times more likely to be stunted compared to children aged below 6 months. Similarly, children aged 24–59 months were exp(1.8917) = 6.6306 (95% CI:5.1732, 8.4977) times more likely to be stunted compared to children aged below 6 months.

The previous birth interval is a significant determinant of childhood stunting. The estimated odds ratio (OR = 0.7945; 95% CI: 0.6523, 0.9676) reveals that children having a birth interval of 48 months or above had 20.6% less risk of being stunted compared to children having a birth interval of less than 24 months. That means that the longer the interval, the less likely it is that the child will be stunted. Generally, the birth interval has an inverse relationship with stunting.

The place of residence is a significant determinant of stunting. Children from rural residence were also 60.43% (OR = 1.6043, 95%CI: 1.2682, 2.0297) times more likely to be stunted compared to children from urban residence after controlling for other predictors in the model. Other predictors can be interpreted in the same way.

The ALR model also presents the estimated constant log odds of alpha, which provides information about the association between individual observations within the same cluster. The estimated pairwise odds ratio relating two responses from the same cluster is exp(0.2114) = 1.2354 (95% CI: 1.1367, 1.3427). Thus, the value of log odds of alpha greater than one indicates that the association is found to be significant (p-value < .0001), and this means that there is a strong positive association between individual children regarding stunting in the same cluster.

## Discussion

This study was intended to determine the prevalence of childhood stunting and identify the risk factors for stunting in children aged five years or younger in Ethiopia based on EDHS

2016 data. Stunting was measured by the height-for-age z-scores. As a preliminary analysis, an assortment of summary statistics was employed to explore the association between the response variable of interest and the available predictors. It should be well known that there is inconsistency in the conclusions drawn from the analysis of various summary statistics, which might be due to the fact that they use varying amounts of information, which determines the power of their inferences. Thus, the analysis was extended to other statistical methods to account for the clustered nature of the correlated observations. The data were then analyzed using marginal models (GEE and ALR).

Two proposed working correlation structures, exchangeable and independence correlation assumptions, were taken for comparison in the GEE model-building strategy. The model with an exchangeable working correlation structure was found to better fit the data independence. This supports the idea that, considering clustering, the nature of the data was essential for the analysis and the dependency of individuals on the given data. In addition, ALR was fitted to simultaneously regress the response variable on predictors as well as the association among responses in terms of pair-wise odds ratio.

Two models from marginal model families were compared to assess which model efficiently explains the relations between response and predictors as well as to evaluate whether considering a pair-wise association is important. After that, the ALR model was selected as the best model, and the model shows that there is a positive pair-wise association between responses. This supports the idea explained by Zeger et al. [23], that alternating logistic regression is reasonably efficient relative to GEE. All fitted models led to the same conclusion: age of the child, previous birth interval, education level of the mother, place of residence, wealth index, and BMI of the mother were found to be significantly associated with stunting, whereas sex of the child, age of the mother at first birth, number of household members, number of antenatal care visits, birth order, source of drinking water, having diarrhea recently, and having fever in the two weeks before the survey were not significantly associated with childhood stunting.

Findings from this study showed that the risk of stunting increased with the age of children. Children whose age group was 18–23 months were at higher risk of stunting as compared to children aged below 6 months. Similar studies were reported in Ethiopia, India, Peru, Vietnam, Bangladesh, Madagascar, and Malawi [24–29]. It could be due to the inappropriate and late introduction of low-nutrition supplementary food [30], and a large portion of guardians in rural areas are ignored to meet their children's optimal food requirements as the child's age increases [31]. This result is also consistent with the 2016 national report of EDHS, which indicated that the prevalence of stunting increases as the age of the child increases [32].

Previous birth intervals showed a highly significant and inverse relationship with the prevalence of stunting. Children with longer previous birth intervals had a lower risk of being stunted. The analysis of this study shows that both birth interval groups (24 to 47 months and 48 months or older) were statistically significant. Thus, the likelihood of being stunted decreases as the birth interval increases. This finding is confirmed by most previous studies [33–36]. The significant and higher risk of stunting among children which a birth interval could be due to uninterrupted pregnancy and breastfeeding, since this drains women's nutritional resources [37]. Close spacing may also have a health effect on the previous child, who may be prematurely weaned if the mother becomes pregnant too early again. In this study, rural children were found to be most affected by stunting. This may be due to close spacing and a low contraceptive prevalence rate in the rural areas. This is inconsistent with the study conducted by [28, 38]. They didn't find birth order to be significantly related to stunting for the whole sample by using multivariable multilevel logistic regression and ordinary logistic regression, respectively.

In this study, the mother's body mass index, which is defined as her weight in kilograms divided by the square of her height in meters, was found to be negatively associated with childhood stunting. Thin mothers (BMI < 18.5) are themselves malnourished and are therefore likely to have stunted children. The same finding is also found in a number of studies. Mothers with low BMI, on average, are giving birth to babies with low birth weights [28, 39, 40]. This finding is also consistent with some of studies in Africa [41, 42]. A mother's nutritional status affects her ability to successfully carry, deliver, and care for her children and is of great concern in its own right. Women who are malnourished (thinness or obesity) may have difficulty during childbirth and may deliver a child who is malnourished. The results indicate that there is an association between the thinness of the mother and the nutritional status of the child.

Childhood stunting was found to be inversely related to the mother's level of education. On average, the risk of stunting was significantly lower for children whose mothers have a secondary or higher level of education than for children whose mothers have no education status. This finding seemed to be consistent with other studies [24, 28, 43, 44]. These findings demonstrate the importance of girls' education as an alternative strategy to beat the burden of childhood stunting and to push sensible feeding practices for young children. Higher levels of maternal education can also reduce childhood stunting in other ways, such as through increased knowledge of sanitation practices and healthy behaviors.

Childhood stunting was also found to be inversely related to the wealth index of households. Children in poor households are found to be, on average, at a higher risk of stunting than children from rich households. This finding was supported by previous literatures [24, 25, 45–47].

As shown in the analysis, urban children were less likely to be stunted (malnourished) than their rural counterparts because the quality of the health environment and sanitation is better in urban areas, whereas the living conditions in rural areas were associated with poor health conditions and a lack of personal hygiene, which were the risk factors in determining malnutrition. This is consistent with some studies, where mothers' place of residence had a statistically significant effect on their children's nutritional status [24, 28, 48].

## Limitations of the study

As a strength, the present study utilized nationwide data to estimate the prevalence and correlates of stunting. In addition, it utilized a strong statistical model. However, being cross-sectional, it could not show the cause-and-effect relationship between stunting and the identified correlates. Moreover, the present study did not include some clinically plausible variables like breastfeeding status, mother's occupation, and exposure to mass media that could have an association with childhood stunting due to the high missing values in the EDHS data.

## Conclusions

GEE and ALR have been compared to analyze the average effect of predictors on the response variable. It is concluded that the ALR model with a measure of association exhibited a better fit for this data than the GEE models. This study suggests that a mother's education level, BMI, previous birth interval, and wealth index adversely affect childhood stunting. In contrast, a child's age and place of residence positively affect childhood stunting. However, in this study, the child's sex, mother's age at first birth, number of household members, number of antenatal care visits, birth order, source of drinking water, diarrhea recently, and fever before two weeks of the survey were not significantly associated with childhood stunting. More importantly, this study contributes to the understanding of the individual and collective effects of maternal, socio-economic, and child-related risk factors of childhood stunting in Ethiopia.

## Supporting information

**S1 Checklist. StroBE checklist.**
(DOCX)

**S1 Data. The 2016 EDHS data.**
(XLSX)

## Acknowledgments

We acknowledge the Ethiopian Central Statistical Agency for providing us the data.

## Author Contributions

**Conceptualization:** Woldemariam Erkalo Gobena, Teramaj Wongel Wotale, Wubishet Gezimu.

**Data curation:** Woldemariam Erkalo Gobena, Teramaj Wongel Wotale, Mesfin Esayas Lelisho, Wubishet Gezimu.

**Formal analysis:** Woldemariam Erkalo Gobena, Teramaj Wongel Wotale, Mesfin Esayas Lelisho.

**Funding acquisition:** Teramaj Wongel Wotale.

**Investigation:** Teramaj Wongel Wotale.

**Methodology:** Woldemariam Erkalo Gobena, Teramaj Wongel Wotale, Mesfin Esayas Lelisho, Wubishet Gezimu.

**Project administration:** Woldemariam Erkalo Gobena.

**Resources:** Woldemariam Erkalo Gobena, Teramaj Wongel Wotale, Wubishet Gezimu.

**Software:** Woldemariam Erkalo Gobena, Teramaj Wongel Wotale, Mesfin Esayas Lelisho.

**Supervision:** Mesfin Esayas Lelisho, Wubishet Gezimu.

**Validation:** Woldemariam Erkalo Gobena, Teramaj Wongel Wotale.

**Visualization:** Woldemariam Erkalo Gobena, Teramaj Wongel Wotale.

**Writing – original draft:** Woldemariam Erkalo Gobena, Teramaj Wongel Wotale, Mesfin Esayas Lelisho, Wubishet Gezimu.

**Writing – review & editing:** Woldemariam Erkalo Gobena, Teramaj Wongel Wotale, Mesfin Esayas Lelisho, Wubishet Gezimu.

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
