## [Decision Letter · Decision Letter 0]

30 Aug 2023

PONE-D-23-15241Prevalence and correlates of stunting among under-five children in Ethiopia:  Marginal models analysis of 2016 Ethiopian demographic and health survey dataPLOS ONE

Dear Dr. Gezimu,

Thank you for submitting your manuscript to PLOS ONE. After careful consideration, we feel that it has merit but does not fully meet PLOS ONE’s publication criteria as it currently stands. Therefore, we invite you to submit a revised version of the manuscript that addresses the points raised during the review process. Please submit your revised manuscript by Oct 14 2023 11:59PM. If you will need more time than this to complete your revisions, please reply to this message or contact the journal office at plosone@plos.org. Please include the following items when submitting your revised manuscript:A rebuttal letter that responds to each point raised by the academic editor and reviewer(s). You should upload this letter as a separate file labeled 'Response to Reviewers'.A marked-up copy of your manuscript that highlights changes made to the original version. You should upload this as a separate file labeled 'Revised Manuscript with Track Changes'.An unmarked version of your revised paper without tracked changes. You should upload this as a separate file labeled 'Manuscript'.

We look forward to receiving your revised manuscript.

Kind regards,

Gizachew Gobebo Mekebo

Academic Editor

PLOS ONE

Journal Requirements:

https://globalizationandhealth.biomedcentral.com/articles/10.1186/s12992-019-0505-7

https://journals.plos.org/plosone/article?id=10.1371%2Fjournal.pone.0256722

https://repository.ju.edu.et/bitstream/handle/123456789/3470/dani%20thesis%20%20to%20be%20printed.pdf?isAllowed=y&sequence=1

In your revision ensure you cite all your sources (including your own works), and quote or rephrase any duplicated text outside the methods section. Further consideration is dependent on these concerns being addressed.

4. We note you have included a table to which you do not refer in the text of your manuscript. Please ensure that you refer to Table 5 in your text; if accepted, production will need this reference to link the reader to the Table.

Reviewers' comments:

Reviewer's Responses to Questions

**Comments to the Author**

1. Is the manuscript technically sound, and do the data support the conclusions?

Reviewer #1: Yes

Reviewer #2: Yes

2. Has the statistical analysis been performed appropriately and rigorously? 

Reviewer #1: Yes

Reviewer #2: No

3. Have the authors made all data underlying the findings in their manuscript fully available?

Reviewer #1: No

Reviewer #2: Yes

4. Is the manuscript presented in an intelligible fashion and written in standard English?

Reviewer #1: No

Reviewer #2: Yes

5. Review Comments to the Author

Reviewer #1: Comments to the Author

Overall

The manuscript presents an important issue, to examine Prevalence and correlates of stunting among under-five children in Ethiopia: The author used 2016 EDHS data to examine the status and correlates of stunting in Ethiopia and used Marginal

models to analysis the data. The 2016 EDHS data source and title that the authors used for this study were outdated, and several scholars have conducted studies on the same title from various perspectives and using various models. However, the authors need to adequately describe the details in some sections. Please look into the following comments in detail.

Abstract

1. Under the conclusion section of the abstract (line 33), the author suggested that to mitigate/reduce childhood malnutrition better to work on improving the knowledge and practice of parents on appropriate young child feeding practices, frequent growth monitoring, and appropriate and timely interventions. This conclusion/suggestion is inconsistent with the findings of the study. The authors may reconsider this point in line with the objective and findings of the study.

Introduction

2. The introduction section is mainly focused on malnutrition rather than stunting, the author may add a paragraph about stunting.

Data Source

3. Why did the author prefer to use the outdated 2016 EDHS data when the recent 2019 EDHS data is available?

Statistical analysis

4. The author may clearly specify why and when the two marginal models were used in this study.

Discussion and Conclusion

5. The paper focuses a lot on the different models, but not on the outcome of these tools. If the purpose is to compare models, then perhaps more comparisons of the different models should be presented.

Reviewer #2: Comments and questions for authors

I appreciated your study, even if it needs some editions on the Abstract, introduction, methods, and results parts, and also on some grammatical errors.

1. In the Title: Why do you use "correlates" rather than association?

2. In the Abstract: Write the prevalence of stunting in the result section of the Abstract.

3. In the Introduction:

Please write more about stunting than malnutrition since your title focuses on stunting.

Please correct the following grammatical errors:

On page 3, lines 50 and 51, "cause for almost half" by "of, and "communities of developing countries" by "in." And some others (check others).

The authors stated a regional variation of stunning/malnutrition in Ethiopia on page 3. Still, the author didn't include the region type as a single variable or not use a multilevel logistic model to examine regional variation as other previous studies found these stunting regional variations (as an example, you can look your reference # 19 conducted by W. Mekonnen Haileselassie et al., 2019).

And, if you look at this interesting article, I don't know the importance of this study (your study). You may need to apply the Generalized Estimating Equations and alternating logistic regression models even if it is not recommended to use them for this DHS dataset (read the Note on the methods and materials section)!

The authors have not stated clearly the gaps (ways of sampling design and clustering) with many previous related studies in Ethiopia. So please state the holes clearly in this section.

4. In the Methods and Materials:

Please write the sampling design and clustering methods clearly since they are the gaps with other previous studies, as you said on page 4.

Why did you use 626 clusters since there are 643 clusters in the 2016 EDHS data?

I don't know why you applied the Generalized Estimating Equations and alternating logistic regression models for this single DHS data. The data is neither Longitudinal (repeated) nor correlated/clustered data!

Note! (1) The Generalized Estimating Equations (GEE) procedure extends the generalized linear model to allow for analysis of repeated measurements or other correlated observations, such as clustered data! (2) The GEE method was developed by Liang and Zeger (1986) to produce regression estimates when analyzing repeated measures with non-normal response variables. The response variable (Y) can be either categorical or continuous. (3) The alternating logistic regressions (ALR) algorithm of Carey, Zeger, and Diggle (1993) models the association between pairs of responses by using log odds ratios instead of using correlations, as ordinary GEEs do.

So, what are the pairs of responses in this study dataset or 2016 EDHS datasets?

5. In the Results: Please correct the following grammatical errors:

On lines 146 and 148, remove "who were" and "of" in "children who were included" and "male and of female".

6. In the conclusion: it needs more editions! So please use the edited conclusion as follows "GEE and ALR have been compared to analyze the average effect of predictors on the response variable. It is concluded that the ALR model with a measure of association exhibited a better fit for this data than the GEE models. This study suggests that a mother's education level, BMI, previous birth interval, and wealth index adversely affect childhood stunting. In contrast, the child's age and place of residence positively affect childhood stunting. However, in this study, the child's sex, mother's age at first birth, number of household members, number of antenatal care visits, birth order, source of drinking water, diarrhea recently and fever before two weeks of the survey were not significantly associated with childhood stunting. More importantly, this study contributes to understanding the individual and collective effect of maternal, socio-economic and child-related risk factors of childhood stunting in Ethiopia".

7. In the Discussions: it also needs more grammatical editions!

Finally, please write the limitations of the study!

6. PLOS authors have the option to publish the peer review history of their article (what does this mean?). If published, this will include your full peer review and any attached files.

Reviewer #1: No

Reviewer #2: **Yes: **Alemayehu Siffir Argawu (PH.D. Scholar in Statistics)

---

## [Author Response · Author response to Decision Letter 0]

19 Sep 2023

Response to reviewers

Manuscript ID: PONE-D-23-15241

Research Title: Prevalence and correlates of stunting among under-five children in Ethiopia: Marginal models analysis of 2016 Ethiopian demographic and health survey data

To: Editor

Re: Response to reviewers

Dear Editor, 

Thank you for allowing resubmission of our manuscript, with an opportunity to address the reviewers’ comments.

We, authors, would also like to express our gratitude to the anonymous reviewers for carefully reviewing the manuscript and for many thoughtful comments, which can enhance the readability and quality of this manuscript. 

We are uploading (i) our point-by-point response to the comments (below) (response to reviewers), and (ii) an updated manuscript with changes being highlighted by yellow colour.

Best regards,

Wubishet Gezimu, corresponding author of the manuscript

Reviewer #1:

Dear reviewer, thank you for reviewing and forwarding such important comments that enhance the readability and the quality of our manuscript. For easy check-up, our responses and all the changes in the manuscript are highlighted by blue colour. 

 Under the conclusion section of the abstract (line 33), the author suggested that to mitigate/reduce childhood malnutrition better to work on improving the knowledge and practice of parents on appropriate young child feeding practices, frequent growth monitoring, and appropriate and timely interventions. This conclusion/suggestion is inconsistent with the findings of the study. The authors may reconsider this point in line with the objective and findings of the study.

 Authors’ Response: We made correction.

 The introduction section is mainly focused on malnutrition rather than stunting, the author may add a paragraph about stunting. 

 Authors’ Response: We have added a paragraph about stunting. 

 Why did the author prefer to use the outdated 2016 EDHS data when the recent 2019 EDHS data is available?

Authors’ Response: We preferred to use 2016 EDHS data because the recent Mini 2019 EDHS data has many missing observations on some variables. 

 The author may clearly specify why and when the two marginal models were used in this study.

Authors’ Response: Marginal models such as generalized estimating equations (GEE) and alternating logistic regression model (ALR) can be used for the data which has clustering nature. The main goal of the marginal models is to analyze the population-averaged effects of the given predictors on the binary response variable of interest. 

 The paper focuses a lot on the different models, but not on the outcome of these tools. If the purpose is to compare models, then perhaps more comparisons of the different models should be presented.

Authors’ Response: The study has compared the two models (GEE and ALR) and selected the better model to analyse data of childhood stunting. The outcomes/ results were based on the chosen model. On model comparison, we have used quasi-likelihood information criteria (QIC). The model which has the smaller QIC is chosen as the better model to analyse data. Thus, the study focuses both on models and outcome of the tools. 

Reviewer #2:

Dear reviewer, we are so much grateful for your very crucial and constructive comments that help to improve the quality of the manuscript. Your comments are so logical and very important. For easy check-up, our responses and all the changes in the manuscript are highlighted by blue colour. 

 In the Title: Why do you use "correlates" rather than association? 

Authors’ Response: We have corrected it. We have said, “Prevalence and associated factors of stunting among under-five children in Ethiopia: Application of Marginal models using 2016 EDHS data”.

 In the Abstract: Write the prevalence of stunting in the result section of the Abstract.

Authors’ Response: We made correction. We have added prevalence of stunting in the result section of the abstract. 

 In the Introduction:

Please write more about stunting than malnutrition since your title focuses on stunting.

Authors’ Response: We have added a paragraph about stunting in the introduction part of the manuscript. 

Please correct the following grammatical errors:

On page 3, lines 50 and 51, "cause for almost half" by "of, and "communities of developing countries" by "in." And some others (check others).

Authors’ Response: We have corrected grammatical errors which were on page 3, line 50 and 51. 

The authors stated a regional variation of stunning/malnutrition in Ethiopia on page 3. Still, the author didn't include the region type as a single variable or not use a multilevel logistic model to examine regional variation as other previous studies found these stunting regional variations (as an example, you can look your reference # 19 conducted by W. Mekonnen Haileselassie et al., 2019).

Authors’ Response: We made correction on regional variation of stunting. It’s just to state the prevalence of stunting in the regions. 

And, if you look at this interesting article, I don't know the importance of this study (your study). You may need to apply the Generalized Estimating Equations and alternating logistic regression models even if it is not recommended to use them for this DHS dataset (read the Note on the methods and materials section)!

Authors’ Response: Marginal models such as generalized estimating equations (GEE) and alternating logistic regression model (ALR) can be used for the data which has clustering nature. The main goal of the marginal models is to analyze the population-averaged effects of the given predictors on the binary response variable of interest. EDHS data is a clustered data because for the purpose of administration in Ethiopia, regions are divided into zones, zones into weredas and weredas into lowest administrative unit or kebele. During the 2007 census, each kebele was sub-divided into census enumeration areas (EAs) or clusters which were convenient for the implementation of the census. Thus, the 2007 census conducted by the Central Statistical Agency (CSA) provided sampling frame from which the 2016 EDHS sample was drawn. Thus, it’s possible to use marginal models such as Generalized Estimating Equations (GEE) and Alternating Logistic Regressions (ALR) for EDHS data. Many articles are published using marginal models for DHS data. 

The authors have not stated clearly the gaps (ways of sampling design and clustering) with many previous related studies in Ethiopia. So please state the holes clearly in this section.

Authors’ Response: We have stated the gaps (ways of sampling design and clustering) related with previous studies in Ethiopia. 

 In the Methods and Materials:

Please write the sampling design and clustering methods clearly since they are the gaps with other previous studies, as you said on page 4.

 Authors’ Response: Sampling design and clustering methods are included in the methods and materials. 

Why did you use 626 clusters since there are 643 clusters in the 2016 EDHS data?

Authors’ Response: We used 626 clusters since there are 643 clusters in the 2016 EDHS data because the remaining 17 clusters had missing values on height-for-age (stunting) records. The complete height-for-age record was collected from 626 clusters. 

I don't know why you applied the Generalized Estimating Equations and alternating logistic regression models for this single DHS data. The data is neither Longitudinal (repeated) nor correlated/clustered data!

Authors’ Response: EDHS data is a clustered data because in Ethiopia for the purpose of administration, regions are divided into zones, zones into weredas and weredas into lowest administrative unit or kebele. During the 2007 census, each kebele was sub-divided into census enumeration areas (EAs) or clusters which were convenient for the implementation of the census. Thus, the 2007 census conducted by the Central Statistical Agency (CSA) provided sampling frame from which the 2016 EDHS sample was drawn. Thus, it’s possible to use marginal models such as Generalized Estimating Equations (GEE) and Alternating Logistic Regressions (ALR). 

Note! (1) The Generalized Estimating Equations (GEE) procedure extends the generalized linear model to allow for analysis of repeated measurements or other correlated observations, such as clustered data! (2) The GEE method was developed by Liang and Zeger (1986) to produce regression estimates when analyzing repeated measures with non-normal response variables. The response variable (Y) can be either categorical or continuous. (3) The alternating logistic regressions (ALR) algorithm of Carey, Zeger, and Diggle (1993) models the association between pairs of responses by using log odds ratios instead of using correlations, as ordinary GEEs do.

So, what are the pairs of responses in this study dataset or 2016 EDHS datasets?

Authors’ Response: ALR measures the association between pairs of responses using log odds ratio i.e. ALR measures the association between two responses/ outcomes using log odds ratio. Assume Y_jk and Y_jl are two responses/ pair of responses where Y_jk represent child k from cluster j and Y_jl represent child l from cluster j. Then, the ALR model measures the association of the two outcomes by 

 logit P(Y_jk=1/〖 Y〗_jl=y_jl)=γ_jkl y_jl+log((π_(jk ) 〖- v〗_jkl)/(〖1-π〗_jk-π_jl+v_jkl )) 

Where γ_jkl be the log odds ratio between outcomes Y_jk & Y_jl and π_jk=P(Y_jk=1) and v_jkl=P(Y_jk=1,Y_jl=1). 

 In the Results: Please correct the following grammatical errors:

On lines 146 and 148, remove "who were" and "of" in "children who were included" and "male and of female".

Authors’ Response: We have corrected the grammatical errors which were existed on line 146 and 148. 

 In the conclusion: it needs more editions! So please use the edited conclusion as follows "GEE and ALR have been compared to analyze the average effect of predictors on the response variable. It is concluded that the ALR model with a measure of association exhibited a better fit for this data than the GEE models. This study suggests that a mother's education level, BMI, previous birth interval, and wealth index adversely affect childhood stunting. In contrast, the child's age and place of residence positively affect childhood stunting. However, in this study, the child's sex, mother's age at first birth, number of household members, number of antenatal care visits, birth order, source of drinking water, diarrhea recently and fever before two weeks of the survey were not significantly associated with childhood stunting. More importantly, this study contributes to understanding the individual and collective effect of maternal, socio-economic and child-related risk factors of childhood stunting in Ethiopia".

Authors’ Response: We have edited the conclusion based on your guidance. 

7. In the Discussions: it also needs more grammatical editions!

Authors’ Response: We have edited/ corrected the grammatical errors which existed in the discussions. 

Finally, please write the limitations of the study!

Authors’ Response: We have added the limitations of the study in the manuscript. 

Journal Requirements:

Authors’ Response: Thank you so much for your suggestion. We ensure that our manuscript meets the all PLOS ONE's style requirements

https://globalizationandhealth.biomedcentral.com/articles/10.1186/s12992-019-0505-7

https://journals.plos.org/plosone/article?id=10.1371%2Fjournal.pone.0256722

https://repository.ju.edu.et/bitstream/handle/123456789/3470/dani%20thesis%20%20to%20be%20printed.pdf?isAllowed=y&sequence=1

In your revision ensure you cite all your sources (including your own works), and quote or rephrase any duplicated text outside the methods section. Further consideration is dependent on these concerns being addressed.

Authors’ Response: Thanks a lot for this very important suggestion. We have your suggestion and accordingly made a significant changes (rephrased the duplicated texts) to our manuscript. Thanks once again.

Authors’ Response: Thanks so much. We have included the minimal underlying data set in the supporting information of the updated manuscript. Again we have uploaded it.

4. We note you have included a table to which you do not refer in the text of your manuscript. Please ensure that you refer to Table 5 in your text; if accepted, production will need this reference to link the reader to the Table.

Authors’ Response: Thanks a lot. We have corrected (referred table five in text) the mentioned section.

---

## [Decision Letter · Decision Letter 1]

4 Oct 2023

PONE-D-23-15241R1Prevalence and associated factors of stunting among under-five children in Ethiopia:  Application of marginal models analysis of 2016 Ethiopian demographic and health survey dataPLOS ONE

Dear Dr. Gezimu,

Thank you for submitting your manuscript to PLOS ONE. After careful consideration, we feel that it has merit but does not fully meet PLOS ONE’s publication criteria as it currently stands. Therefore, we invite you to submit a revised version of the manuscript that addresses the points raised during the review process. Please submit your revised manuscript by Nov 18 2023 11:59PM. If you will need more time than this to complete your revisions, please reply to this message or contact the journal office at plosone@plos.org. Please include the following items when submitting your revised manuscript:A rebuttal letter that responds to each point raised by the academic editor and reviewer(s). You should upload this letter as a separate file labeled 'Response to Reviewers'.A marked-up copy of your manuscript that highlights changes made to the original version. You should upload this as a separate file labeled 'Revised Manuscript with Track Changes'.An unmarked version of your revised paper without tracked changes. You should upload this as a separate file labeled 'Manuscript'.If applicable, we recommend that you deposit your laboratory protocols in protocols.io to enhance the reproducibility of your results. Protocols.io assigns your protocol its own identifier (DOI) so that it can be cited independently in the future. For instructions see: https://journals.plos.org/plosone/s/submission-guidelines#loc-laboratory-protocols. Additionally, PLOS ONE offers an option for publishing peer-reviewed Lab Protocol articles, which describe protocols hosted on protocols.io. Read more information on sharing protocols at https://plos.org/protocols?utm_medium=editorial-email&utm_source=authorletters&utm_campaign=protocols.

We look forward to receiving your revised manuscript.

Kind regards,

Gizachew Gobebo Mekebo

Academic Editor

PLOS ONE

Journal Requirements:

Reviewers' comments:

Reviewer's Responses to Questions

**Comments to the Author**

1. If the authors have adequately addressed your comments raised in a previous round of review and you feel that this manuscript is now acceptable for publication, you may indicate that here to bypass the “Comments to the Author” section, enter your conflict of interest statement in the “Confidential to Editor” section, and submit your "Accept" recommendation.

Reviewer #1: All comments have been addressed

Reviewer #2: All comments have been addressed

2. Is the manuscript technically sound, and do the data support the conclusions?

Reviewer #1: Yes

Reviewer #2: Yes

3. Has the statistical analysis been performed appropriately and rigorously? 

Reviewer #1: Yes

Reviewer #2: Yes

4. Have the authors made all data underlying the findings in their manuscript fully available?

Reviewer #1: Yes

Reviewer #2: Yes

5. Is the manuscript presented in an intelligible fashion and written in standard English?

Reviewer #1: Yes

Reviewer #2: Yes

6. Review Comments to the Author

Reviewer #1: (No Response)

Reviewer #2: Now, the authors have completely responded all the raised questions and comments. And the manuscript is well edited. Thus, it is accepted to publish in this journal.

7. PLOS authors have the option to publish the peer review history of their article (what does this mean?). If published, this will include your full peer review and any attached files.

Reviewer #1: No

Reviewer #2: **Yes: **Alemayehu Siffir Argawu (PHD Scholar)

---

## [Author Response · Author response to Decision Letter 1]

5 Oct 2023

Dear editor and reviewers, thanks a lot for the time with our manuscript so far. All your constructive comments and suggestions boosted up our manuscript’s scientific quality and readability. We have responded to Reviewer #1’s suggestions as mentioned bellow: 

Reviewer #1: 

The authors responded most of my comments. One more comment for author is that please compare your results in discussion section with that of the previous study done by Wake et.al., (2023)

https://doi.org/10.1186/s13690-023-01090-7

Authors’ response: Thank you so much for the suggestion. We have discussed our findings with the recommended literature and cited it accordingly as highlighted under the discussion section of the file. Thanks once again.

Journal Requirements:

Please review your reference list to ensure that it is complete and correct. If you have cited papers that have been retracted, please include the rationale for doing so in the manuscript text, or remove these references and replace them with relevant current references. Any changes to the reference list should be mentioned in the rebuttal letter that accompanies your revised manuscript. If you need to cite a retracted article, indicate the article’s retracted status in the References list and also include a citation and full reference for the retraction notice

Authors’ response: We have checked and corrected the references.

---

## [Decision Letter · Decision Letter 2]

11 Oct 2023

Prevalence and associated factors of stunting among under-five children in Ethiopia:  Application of marginal models analysis of 2016 Ethiopian demographic and health survey data

PONE-D-23-15241R2

Dear Dr. Gezimu,

We’re pleased to inform you that your manuscript has been judged scientifically suitable for publication and will be formally accepted for publication once it meets all outstanding technical requirements.

Kind regards,

Gizachew Gobebo Mekebo

Academic Editor

PLOS ONE

Additional Editor Comments (optional):

Reviewers' comments:

Reviewer's Responses to Questions

**Comments to the Author**

1. If the authors have adequately addressed your comments raised in a previous round of review and you feel that this manuscript is now acceptable for publication, you may indicate that here to bypass the “Comments to the Author” section, enter your conflict of interest statement in the “Confidential to Editor” section, and submit your "Accept" recommendation.

Reviewer #1: (No Response)

2. Is the manuscript technically sound, and do the data support the conclusions?

Reviewer #1: (No Response)

3. Has the statistical analysis been performed appropriately and rigorously? 

Reviewer #1: (No Response)

4. Have the authors made all data underlying the findings in their manuscript fully available?

Reviewer #1: (No Response)

5. Is the manuscript presented in an intelligible fashion and written in standard English?

Reviewer #1: (No Response)

6. Review Comments to the Author

Reviewer #1: (No Response)

7. PLOS authors have the option to publish the peer review history of their article (what does this mean?). If published, this will include your full peer review and any attached files.

Reviewer #1: No

---

## [Editor Report · Acceptance letter]

20 Oct 2023

PONE-D-23-15241R2 

Prevalence and associated factors of stunting among under-five children in Ethiopia:  Application of marginal models analysis of 2016 Ethiopian demographic and health survey data 

Dear Dr. Gezimu:

I'm pleased to inform you that your manuscript has been deemed suitable for publication in PLOS ONE. Congratulations! Your manuscript is now with our production department. 

Kind regards, 

on behalf of

Assistant Professor Gizachew Gobebo Mekebo 

Academic Editor

PLOS ONE